# Sugar-based synthesis of an enantiomorphically pure zeolite

Andrés Sala[1], José L. Jordá [ID][1], German Sastre [ID][1], Antonio L. Llamas-Saiz [ID][2], Fernando Rey [ID][1] ✉ & Susana Valencia [ID][1] ✉

Zeolites, well-known by their high selectivities in catalytic and separation processes due to their porous nature, play a crucial role in various applications. One significant long-term objective is the synthesis of enantiopure zeolites, potentially enabling enantioselective processes. Earlier attempts result in partial success, yielding some enantiomorphically enriched zeolites. In this study, we introduce a zeolite synthesis approach utilizing chiral organic structure directing agents (ch-OSDAs) derived from sugars, guiding the crystallization process toward achieving enantiomorphically pure S-STW zeolite. The purity of the zeolite is confirmed through extensive analyses of individual crystals using single-crystal X-ray diffraction, extracting Flack parameters and space groups. Theoretical and structural investigations confirm that the sugar-derived ch-OSDA perfectly fits the characteristic helicoidal channel of the zeolite structure, featuring its efficacy in achieving enantiopure zeolites.

The selective synthesis of enantiopure molecular sieves in the field of zeolites and related materials represents a longstanding challenge and holds profound significance[1,2]. Attaining this goal offers the exciting prospect of conducting enantiomeric recognition processes, of importance in separations or catalytic reactions. In comparison to hybrid organic-inorganic enantiopure materials such as MOFs, inorganic molecular sieves present superior thermal and chemical stability, alongside enhanced environmental and health compatibility. These properties highlight their potential for substantial societal impact beyond their intrinsic scientific value.

The earliest report on chiral zeolites in literature was the structure of zeolite Beta, described by Treacy and Newsam as a complex intergrowth of different polymorphs[3,4]. One of them, the polymorph A, exhibits symmetry compatible with a chiral structure showing two possible helical conformations that can be either right- or left-handed (space groups P4$_1$22 and P4$_3$22, respectively)[4]. However, despite the various polymorphs elucidated in these works, only the non-chiral polymorphs B and C have been obtained as pure phases[5,6]. Up to now, the synthesis of pure polymorph A has remained an elusive endeavor[7,8].

Subsequently, a range of chiral zeolite structures has been cataloged in the IZA database[9], including CZP, GOO, OSO, JRY, -ITV,

STW, LTJ[10–17], all crystallizing in any of the 65 Sohncke space groups[18–20]. To date, none of these zeolites have been obtained as enantiomorphically pure materials; rather, they are obtained as racemic mixtures of chiral crystals composed of distinct enantiomorphs (usually denoted as R and S), highlighting the challenge in achieving enantiopure zeolites[2].

Attempts to obtain highly enantioselective synthesis methods include the recrystallization of a racemic mixture of two polymorphs in the presence of a chiral induction agent, able to catalyze the transformation of the racemic mixture of zeolites in one of the pure enantiomorphs. This was achieved by transforming the racemic mixture of a Zn containing aluminophosphate (ZnAlPO) having the CZP structure (equal proportion of P6$_1$22 or P6$_5$22)[11] in presence of a ribonucleotide as asymmetric catalyst able to transform the racemic mixture in an enantioenriched CZP material with the P6$_1$22 space group, being the selectivity of said transformation of 85%[21].

Thus far, all strategies in preferentially obtaining one enantiomer over the other have involved the utilization of chiral organic structure-directing agents and, in a few cases, have led to significant enantiomeric enrichments in synthesized zeolites. This approach was proposed by Newsam and Treacy for obtaining the above-mentioned

---

[1]Instituto de Tecnología Química, Universitat Politècnica de València - Consejo Superior de Investigaciones Científicas (UPV-CSIC), Av. de los Naranjos, s/n, Valencia, Spain. [2]Unidad de Rayos X (Área Infraestructuras Investigación), Universidad de Santiago de Compostela; Edificio CACTUS, Santiago de Compostela, Spain. ✉e-mail: frey@itq.upv.es; svalenci@itq.upv.es

polymorph A of zeolite Beta[3,4] and later attempted by Davis and Lobo with very moderate success[1].

Over 25 years later, Davis's group achieved a breakthrough by reporting the first instance of an enantiomorphically enriched zeolite, specifically enantioenriched STW zeolite, employing chiral diquaternary imidazolium-based OSDAs[22,23]. The determination of zeolite STW's enantiomeric excess was assessed by high-resolution transmission electron microscopy (HRTEM) analysis, albeit on a statistically limited number of individual crystals (6 crystals per sample). Additionally, the flexible conformational points within the ch-OSDA structure allow for the formation of both enantiomers, with a potential bias toward one of them[22].

More recently, Gómez-Hortigüela et al. reported the syntheses of -ITV zeolites using an ephedrine-based ch-OSDA (called GTM-3[24,25] and GTM-4[26]), indicating a significant enantiomorphic excess suggested by catalytic testing for asymmetric epoxide opening (55% e.e.)[26]. However, the obtained solids also contained some amorphous material, and the role of this non-crystalline solid in chiral recognition was not determined[24–26]. This is particularly relevant considering the relatively large size of the reacting epoxides in relation to the pore size of the -ITV zeolite and the high conversions reported in that work. All these results underscore the need for further optimization to achieve enantiopure chiral zeolites.

In this study, we propose the utilization of a sugar-derived compound as a building block for advancing a family of chiral-organic structure-directing agents. Specifically, isomannide, a highly rigid diol formed by two fused five-membered rings derived from D-mannitol[27], is identified as a promising candidate. Isomannide is well-documented for its versatility as a platform chemical, finding applications ranging from polymers to pharmaceuticals[28,29]. An important feature of isomannide is the absence of a specular analog, rendering as a highly appropriate starting molecule for the synthesis of enantiopure ch-OSDAs[27].

## Results and discussion

We synthesized a series of chiral organic dications containing isomannide as chiral spacer, resulting in enantiopure rigid dications that are shown in Fig. 1.

**Fig. 1 | Sugar based organic structure directing agents used in the zeolite synthesis experiments.** Chemical structures of the four different sugar based ch-OSDAs used for studying their ability to synthesize chiral zeolites.

The methodology employed for the synthesis of these ch-OSDAs involved the nucleophilic substitution of both hydroxyl groups, leading to the inversion of the stereochemistry with respect to the original isomannide, as shown in Suppl. Figure 1. The primary role of isomannide-linker is acting as spacer for the positively charged nitrogen, while its intrinsic restricted conformation also induces a spatial asymmetry in these dicationic chiral compounds. By applying this approach, a family of ch-OSDAs with variable chemical nature was obtained. The detailed synthesis procedures of each ch-OSDA preparation is reported in Methods "section".

The ch-OSDA cations were essayed in zeolite synthesis conditions and it was found that all of them were able to direct the crystallization of different zeolites. The main synthesis conditions for the zeolites are given in Table 1, along with typical X-ray diffraction (XRD) patterns of the obtained zeolites, which are shown in Suppl. Figure 2.

Our study revealed that the ch-OSDA3 was highly selective for the synthesis of zeolite STW in the presence of Ge and in fluoride media. The observed selectivity of ch-OSDA3 towards zeolite STW was rationalized by means of theoretical calculations (see computational details in Methods "section"). The calculated van der Waals energies of the different ch-OSDAs within the STW structure show that only ch-OSDA3 is stabilized in the micropore of S-STW (P6₁22 space group), while the others result in very high interaction energies precluding the formation of zeolite STW (Suppl. Table 1). These results can be easily visualized in Suppl. Figure 3 shows OSDA atoms protruding outside the STW micropore, leading to repulsion, and indicating clearly that ch-OSDA3 is the molecule with less repulsive interactions. Furthermore, theoretical calculations elucidating the Van der Waals interaction between the R-STW framework and the ch-OSDA3 suggest that its location within the pores of the R-STW enantiomorph is significantly less favorable compared to the S-STW zeolite, as shown in Suppl. Figure 4.

Additionally, other zeolites have been obtained with this family of ch-OSDAs, among them Beta and BEC. None of the studied organic compounds favored the preferential crystallization of polymorph A over other polymorphs of zeolite Beta, highlighting the evidence that chiral OSDAs may not transmit chiral properties to the final zeolite and thus, non-chiral zeolites can crystallize with chiral OSDAs, as was observed previously using enantiopure amino acid derived-OSDAs[30]. Consequently, our attention was focused on STW zeolites since, as mentioned earlier, this zeolite possesses a chiral structure.

Zeolite STW was obtained as silicogermanate with Si/Ge ratios ranging from 1 to 5 (Suppl. Fig. 5). Attempts to synthesize zeolite STW without Ge were unsuccessful, leading to the formation of amorphous solids (Table 1). Equivalent results were achieved when employing Ge-containing STW zeolite (2-STW sample, Suppl. Table 2) as seeds.

The integrity of ch-OSDA3 occluded in zeolites STW was confirmed by ¹³C-CP-SS-NMR spectroscopy (Suppl. Fig. 6). The NMR spectrum of the occluded cation was very similar to that of the free dication, indicating that the chiral OSDA was incorporated into the solid without any evident decomposition.

**Table 1 | Synthesis conditions used in selected experiments and phases obtained using different organic structure direct agents in fluoride medium**

| T (°C) | t (days) | Composition | OSDA1 | OSDA2 | OSDA3 | OSDA4 |
|--------|----------|-------------|----------|-------|-------|-------|
| 175 | 7–14 | Si | MTN, AST | Beta | Am | Am |
| 150 | 3–7 | Si/Ge = 2 | BEC | BEC | STW | ITQ-21 |
| 150–175 | 14–20 | Si/Al = 25 | Am | Beta | Am | Am |
| 175 | 10–20 | Si/Al = 50 | Am | - | Am | Am |
| 175 | 10–20 | Si/B = 15 | - | Beta | Am | - |

*Am* amorphous material.

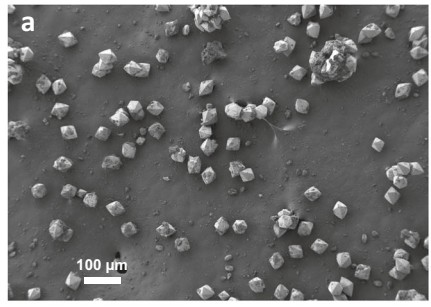
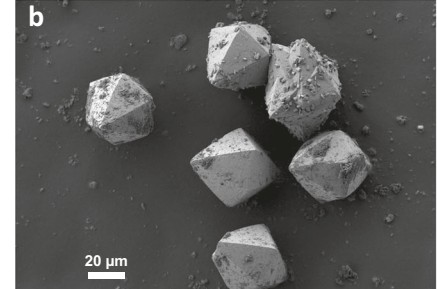
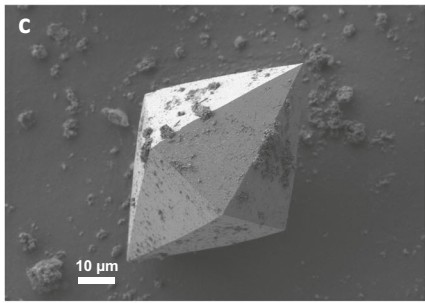
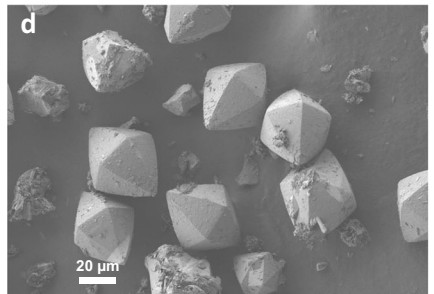

**Fig. 2 | Field emission scanning electron microscopy images of three different preparations of enantiomorphically pure 2-STW zeolite in the as-made form.** The images show the typical shape of STW crystals and prove the homogeneity in the crystal size distribution and the high reproducibility of the synthesis. **a, b** FE-SEM images of sample 2-STWa at different magnifications, **c** FE-SEM image of sample 2-STWb, and (**d**) FE-SEM image of sample 2-STW-25, samples labeled as in Suppl. Table 4.

Additionally, the stability of the zeolite STW was investigated after the removal of the OSDA through calcination. The XRD patterns of the in-situ calcined sample of Si/Ge ratio close to 1 at different temperatures are shown in Suppl. Fig. 7, revealing that the crystallinity remained intact up to temperatures as high as 600 °C. The occluded OSDA was removed at around 500 °C, as indicated by the increase in intensities of the low-angle XRD peaks. The XRD pattern of the calcined material measured at room temperature avoiding its contact with air shows that crystallinity was completely retained (Suppl. Fig. 8). However, the calcined STW sample significantly reduces its crystallinity after exposition to atmospheric moisture, resulting in an amorphous material upon prolonged contact with air.

Textural properties were determined from $N_2$ and Ar isotherms of the calcined zeolite STW, avoiding its contact with atmospheric moisture. The apparent BET area and micropore volume of the Ge-containing zeolite STW (sample 2-STW of Suppl. Table 3) were lower than those of the pure silica analog due to the presence of high content of Ge replacing Si in the STW framework. Thus, we calculated the hypothetical micropore volume and apparent BET area of sample 2-STW as pure silica (sample Si-STW in Suppl. Table 3). The recalculated values were very similar to a pure silica STW zeolite synthesized following a procedure described in literature[31] (sample HPM-1 of Suppl. Table 3). The micropore distribution obtained from the Ar isotherm of sample 2-STW was centered on 5.0 Å, in good agreement with the pore aperture from the crystallographic structure of STW (assuming O diameter of 2.5 Å). Textural properties and Ar isotherm are given as supplementary information (Suppl. Table 3 and Suppl. Fig. 9).

The main objective of using chiral OSDAs in the synthesis of zeolites is to obtain enantiopure zeolites. Unfortunately, there is a lack of appropriate characterization methods for assessing the handedness of nanometric crystals, typically obtained in zeolite syntheses. Advanced electron microscopy techniques[32–34] and circular birefringence of the material[35] have been used for assessing the absolute structure of individual chiral crystals of a zeolitic material. However, these techniques were applied to racemic mixtures or enantiomorphically enriched zeolites. Nevertheless, the most widely accepted

method for confirming the enantiopurity of crystalline materials is the refined Flack parameter (*x*) obtained from single-crystal X-ray diffraction (SCXRD) data. If the refinement gives $x \leq 0.10$, the absolute configuration is confirmed[18,19]. Therefore, to obtain an unambiguous extrapolation of the chiral sense to the entire sample, a statistically significant number of crystals randomly selected from the bulk must be determined through a single-crystal analysis.

Optical microscopy was utilized to study the ch-OSDA-STW material showing the presence of extinction orientations at 0° and 90° with respect to the plane of the polarized light, while crystals oriented at 45° and 135° exhibited high brilliance (Suppl. Fig. 10). This result suggests the presence of chiral crystals in the studied sample, although it cannot be ruled out that the observed optical behavior was due to the presence of chiral organic moieties inside the zeolite STW.

Thus, we took advantage of the very large crystals of zeolite STW of approximately 70×50 μm² evidenced by Field Emission Scanning Electron Microscopy (FE-SEM) (images shown in Fig. 2) for carrying out a detailed single crystal XRD study of some selected STW samples.

Consequently, 30 different single crystals were selected from three different synthesis batches, with two sets of samples being silico-germanates and the third material, a silico-alumino-germanate. The absolute configurations of these 30 crystals were individually assessed by determining their Flack parameters (Suppl. Table 4). All the crystals were indexed in the space group $P6_122$ (enantiomorph S) and displayed Flack parameters closer to 0 and less than 3 times their standard uncertainties (s.u.)[18,19]. The statistical analysis reveals that the independent examination of 30 single crystals, all exhibiting identical chirality, permits a confident inference regarding sample purity, indicating a minimum purity level of 95% with a statistical certainty exceeding 80%, but with no evidence of the presence of the enantiomorph R.

For the sake of comparison, the Flack parameters of 11 different single-crystals of a Ge-containing STW sample (FE-SEM images shown in Suppl. Fig. 11) of similar Ge content, but synthesized using a non-chiral OSDA[36] were determined (Suppl. Table 5). In this case, six crystals were indexed in the space group $P6_522$ (enantiomorph R) and five

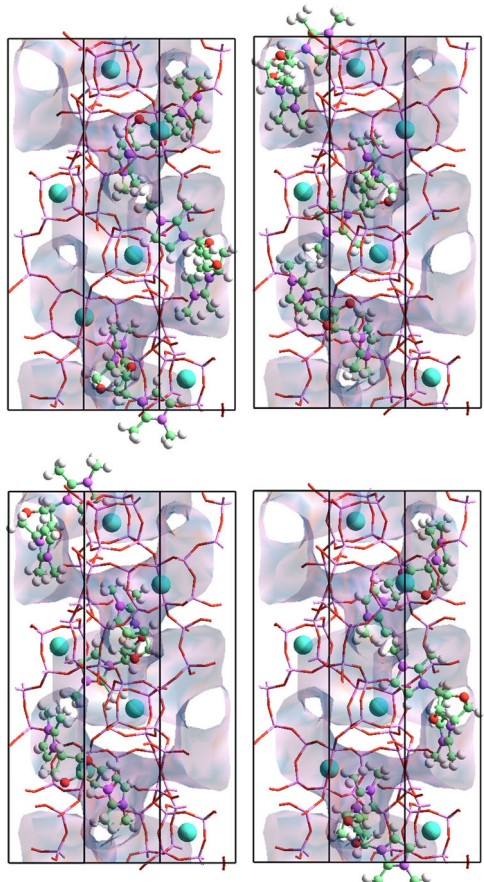

**Fig. 3 | View of the four possible arrangements of OSDA3 molecules filling the STW channels.** View along axis [110] of the $1 \times 1 \times 1$ STW unit cell showing the four ordering of the OSDA molecules located inside the helicoidal channels of the framework. Structural refinement shows that each OSDA arrangement has exactly the same probability of occupying the zeolite microporosity. The atom color code is as follows: For the OSDA: red: oxygen; green: carbon; pink: nitrogen, hydrogen is not shown for clarity. Cyan: Fluorine atoms located at the D4R units. STW wireframe: Light pink stick: silicon, red stick: oxygen.

in the space group $P6_122$ (enantiomorph S) indicating that a racemic mixture of both enantiomorphs was formed.

All these findings provide clear evidence that the utilization of a sugar-based synthesis strategy has resulted in the selective production of an enantiomorphically pure zeolite, specifically the S-STW, which has been searched for a long time and never found until this report.

The complete structure refinement of the enantiopure zeolite S-STW from single crystal X-ray diffraction data confirmed that the stereochemistry of the OSDA (R, R) is preserved during crystallization and provides useful information about its location and conformation (absolute configuration). The unit cell formula determined was $C_{48}H_{72}N_{12}F_6Si_{28.7}Ge_{31.3}O_{126}$. The main crystallographic data and further details on data acquisition are shown in supplementary information (Suppl. Table 6).

The structure of S-STW possesses a helicoidal channel that hosts three ch-OSDA molecules per unit cell, allowing the two charged imidazolium groups of each molecule to be positioned at a 60° angle to each other, and restricting their intramolecular mobility. In this conformation, the distance between the positive charges of the occluded organic matches the spacing of adjacent units $[4^65^88^210^2]$ that define the helical pore of the STW structure (Fig. 3). The methyl substituents on the imidazolium sterically fill the volume of the cavity, and they are an important factor in the stabilization of these units.

Detailed refinement of the structure indicates the presence of four possible locations for the OSDA molecules inside the channels, with an equal probability of ¼ for each one (Fig. 3). The arrangement of the occluded molecules strictly follows the ordering of the $[4^65^88^210^2]$ units along the channel direction, but only one of these four possible distributions can appear in each channel. A theoretical overlap of all four molecules describes a helix, filling completely the channels of the STW structure with the same pitch and handedness. Thus, the result is the selective crystallization of only one of the two possible enantiomorphs of STW, in particular S-STW.

The purity of the S-STW sample was confirmed by carrying out the Rietveld refinement of the powder X-ray diffraction pattern of the as-made material. The experimental diffractograms were refined using the FullProf program, confirming the purity of the analyzed zeolite and the fact that the selected crystals were fully representative of the whole synthesized zeolite STW (Suppl. Table 7 and Fig. 4a). In addition, the zeolite S-STW was calcined in dry air, and its X-ray diffraction pattern was refined also using the Rietveld method. The results show that the calcined material retains the structural features observed in the as-made solid with a very slight modification of the unit cell parameters (Suppl. Table 8 and Fig. 4b).

The refinement of the T-sites occupancies indicates that the Si/Ge ratio of the analyzed single crystal of STW is 0.92, which agrees with the chemical analysis of the powder sample (Si/Ge=1.3). The preferential occupancy of Ge in sites T1 and T2 (Si/Ge = 0.54) is clear, being less evident for T3 and T4 (Si/Ge = 0.85 and 0.82, respectively), while T5 site (Si/Ge = 4.88) contains a much higher proportion of Si. Thus, T1, T2, T3, and T4, which belong to D4R units, are Ge enriched, whilst T5 position, neighboring D4R units, is strongly enriched in Si. The preferential occupancy of Ge in the D4R cages of the STW structure is proved, as commonly observed for other zeolitic silicogermanates[37–39].

Since the formation of enantiomorphically pure S-STW has been demonstrated, its potential in enantioselective applications, as described by Brand[22] and de la Serna[24–26], has been pursued. However, the significant diffusional limitations arising from the large crystal size of the synthesized materials in this study pose a challenge. To address this issue, attempts were made to reduce the crystal size. Various synthesis parameters were adjusted, and seed crystals were introduced into the synthesis gels in an effort to decrease the average crystal size of the S-STW material. Despite these attempts, no substantial reduction in the crystal size of zeolite S-STW was observed when crystallized as a pure phase. Consequently, both the catalytic activity observed in epoxide ring-opening reactions and the adsorption capacity of chiral alcohols on the zeolites S-STW reported herein were low. Thus, further research is needed to overcome this limitation, or it may open possibilities for other applications utilizing enantiomorphically pure zeolites, such as the one described in this report.

The synthesis of enantiomorphically pure zeolites has been demonstrated through an innovative synthesis strategy employing sugar-derived organic structure directing agents. These findings serve as a convincing proof of concept and unambiguously show the feasibility of obtaining enantiomorphically pure S-STW zeolites.

## Methods
### Synthesis of the chiral organic structure directing agents
The general synthetic route followed to obtain the organic structure directing agents derived from isomannide (Fig. 1) is summarized in Suppl. Figure 1.

Synthesis of cations derived from 1,4:3,6-dianhydro-D-mannitol (isomannide) have been carried out following two different methods depending on the type of functionalization groups. The synthesized compounds share a first stage of activation of the alcohol group to form product 1 (Iso-Ts) or product 2 (Iso-Tf), followed by a second step consisting of a bimolecular nucleophilic substitution (SN2). The nature of the nucleophile in this second step determines the type of activation

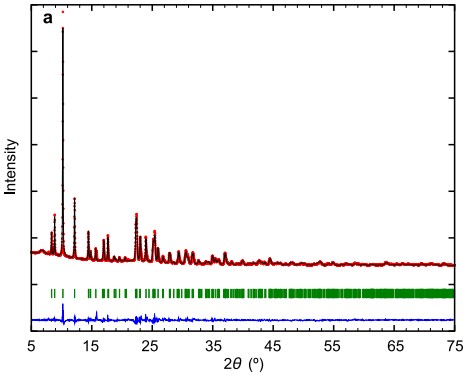

**Fig. 4 | PXRD analysis.** Rietveld refinements of as-made (**a**) and dry-air calcined (**b**) 2-STW samples. Red dots: experimental points; black line: calculated profile; green marks: position of Bragg reflections; blue line: difference pattern. The calcined sample must be kept under dry conditions to avoid its decomposition when exposed to atmospheric moisture. Source data are provided as a Source Data file.

required for isomannide. The syntheses of the different intermediates and final organic products prepared are described below.

**1,4:3,6-dianhydro-2,5-di-O-p-tosyl-D-mannitol (Iso-Ts).** 190 g (1 mol) of $p$-toluenesulfonyl chloride were slowly added to a solution of isomannide (50 g, 0.335 mol) in a mixture of pyridine (90 mL, 1.1 mol) and $CH_2Cl_2$ (250 mL) which had been cooled in an ice bath. The reaction mixture was then poured onto ice water, washed several times with HCl (1 M, 3 × 150 mL) and extracted with $CHCl_3$. The solvent was then removed under reduced pressure, and the resulting precipitate was recrystallized (EtOH/$CHCl_3$) to obtain Iso-Ts as white crystals (145 g, 96%).

$^1$H-NMR (CDCl$_3$, ppm): $\delta$ = 7.78 (d, 4H), 7.32 (d, 4H), 4.83 (q, 2H), 4.46 (dd, 2H), 3.89(dd, 2H), 3.72 (dd, 2H), 2.46 (s, 6H). $^{13}$C-NMR (CDCl$_3$): $\delta$ = 145.38, 133.10, 130.00, 128.02, 80.02, 77.97, 70.16 and 21.68. Theoretical elemental analysis. $C_{30}H_{38}N_4O_8S_2$ (%): C 52.7, N 9.7, H 5.9, S 9.9; exp. C 49.1, N 7.8, H 8.1.

**1,4:3,6-dianhydro-2,5-di-O-trifluoromethanesulfate-D-mannitol (Iso-Tf).** Isomannide (20 g, 134 mmol) was dissolved under $N_2$ in a mixture of anhydrous pyridine (36 mL, 440 mmol) and anhydrous $CH_2Cl_2$ (100 mL) and cooled to 0 °C in an ice bath. Trifluoromethanesulfonic anhydride (72 mL, 400 mmol) was then added dropwise over 1 h and the reaction mixture was kept at room temperature for 12 h. The reaction mixture was then poured onto ice water, washed several times with HCl (1 M, 3 × 100 mL) and extracted with $CHCl_3$. The final product (Iso-Tf) was obtained by recrystallization (EtOH/$CHCl_3$) as white crystals (53.8 g, 98%).

$^1$H-NMR (CDCl$_3$, ppm): $\delta$ = 5.21 (q, 2H), 4.76 (dd, 2H), 4.14(dd, 4H). $^{13}$C-NMR (CDCl$_3$): $\delta$ = 119.34, 84.30, 81.20, 73.14.

The organic cations used as structure directing agents in the synthesis of zeolites (ch-OSDA 1 to ch-OSDA4) are depicted in Fig. 1 and their detailed preparation procedure is described below.

**ch-OSDA1: 1,4:3,6-dianhydro-2,5-di-oxy-bistrimethylammonium as triflate salt.** Trimethylamine (33% w/w EtOH, 77 g, 250 mmol) was added to the Iso-Tf compound (20 g, 49 mmol). The reaction mixture was kept at 40 °C under vigorous stirring for 48 h and then was cooled to room temperature. The solvent and the excess of amine were removed by distillation under reduced pressure. The final product ch-OSDA1 was obtained as a white solid (22.5 g, 87%).

$^1$H-NMR (D$_2$O, ppm): $\delta$ = 5.40 (s, 2H), 4.42 (d, 4H), 4.26 (q, 2H), 3.27 (s, 18H). $^{13}$C-NMR (D$_2$O, ppm): $\delta$ = 126.03, 121.83, 117.62, 113.27, 82.02, 77.79, 66.45, 52.57.

**ch-OSDA2: 1,4:3,6-dianhydro-2,5-di-oxy-bisquinuclidinium as triflate salt.** Quinuclidine (14 g, 123 mol) was added to a solution of Iso-Tf (20 g, 49 mmol) in $CH_3CN$ (50 mL). The mixture was stirred at 50 °C for 72 h. Then, the solvent was removed under reduced pressure and the residue formed was dissolved in $CHCl_3$ and extracted with milliQ $H_2O$ (3 × 50 mL). Finally, the solvent was removed under reduced pressure and the product ch-OSDA2 was obtained as a white precipitate (17.21 g, 78%).

$^1$H-NMR (300 MHz, D$_2$O, ppm) $\delta$ = 5.35 (s,2H); 4.34(m, 4H); 4.01 (m, 2H), 3.54 (m, 12H), 2.23 (m, 2H). 2.02 (m, 12H). $^{13}$C-NMR (75 MHz, D$_2$O, ppm) $\delta$ = 81.48, 76.55, 65.72, 54.24, 22.96, 18.93. Theoretical elemental analysis. $C_{20}H_{30}F_6N_2O_8S_2$ (%): C 39.7, H 5.0, N 4.6, S 10.6; exp. C 41.0, H 5.39, N 4.6, S 6.3.

**ch-OSDA3: 1,4:3,6-dianhydro-2,5-di-oxy-bisimidazole as tosylate salt.** Dimethylimidazole (24 g, 245 mmol) was added on the Iso-Ts compound (30 g, 66 mmol). The mixture was kept at 130 °C under vigorous stirring for 48 h under inert atmosphere. The reaction mixture was then cooled to room temperature and the excess of imidazole was removed by distillation under reduced pressure at 130 °C and subsequently washing with ethyl acetate and diethyl ether. The final product ch-OSDA3 was obtained as a brown, highly viscous ionic liquid (32.4 g, 76%).

$^1$H-NMR (300 MHz, D$_2$O, ppm): $\delta$ = 7.67 (d, 4H), 7.33 (d, 4H), 5.19 (q, 2H), 5.04 (dd, 2H), 4.4(m, 4H), 3.81 (s, 6H), 2.68 (s, 6H), 2.40 (s, 6H). $^{13}$C-NMR (75.5 MHz, D$_2$O, ppm): $\delta$ = 129.42, 125.14, 123.57, 117.53, 86.90, 71.57, 62.98, 34.80, 20.55, 9.22. Theoretical elemental analysis. $C_{30}H_{39}N_4O_8S_2$ (%): C 55.7, H 5.9, N 8.7, S 9.9; exp: C 55.5, N 8.0, H 5.9 S 9.7.

**ch-OSDA4: 1,4:3,6-dianhydro-2,5-di-oxy-bistriethylphosphonium as triflate salt.** To a solution of Iso-Tf (7 g, 17 mmol) in anhydrous toluene (20 mL) cooled to 0 °C (ice bath), triethylphosphine (5 g, 42 mmol) dissolved in toluene (5 mL) was added dropwise under $N_2$ atmosphere. The mixture was stirred vigorously at 100 °C for 72 h. The reaction mixture was then cooled to room temperature, and the formation of two phases was observed. Maintaining anhydrous conditions, the excess toluene was decanted, washed again with anhydrous toluene (3 × 5 mL), ethyl acetate (20 mL), and diethyl ether (20 mL), and dried under reduced pressure. The final product ch-OSDA4 was obtained as a white solid (10.6 g).

$^1$H-NMR (D$_2$O, ppm): $\delta$ = 5.16 (m, 2H), 4.47 (m, 2H), 4.15 (m, 2H), 2.38 (m, 12H), 1.30 (t, 18H) $^{13}$C-NMR (D$_2$O, ppm): $\delta$ = 82.20, 66.78, 37.26, 36.61, 11.26, 10.62, 4.86. $^{31}$P-NMR (D$_2$O, ppm): $\delta$ = 39.28.

### Synthesis of zeolites

The organic structure directing agents used in the zeolite synthesis were converted to their hydroxide forms through ionic exchange with an anion exchange resin (Amberlite IRN-78), according to the following procedure:

Amberlite resin (exchange capacity: 1.1 equivalent per liter) was introduced into a chromatographic column and washed with distilled water until a neutral pH was achieved. Next, a salt of the organic cation to be exchanged (approx. 1 g of solid/10 g of water) was dissolved in water, and the solution was poured onto the column. The solution was then passed through the column three more times, without allowing the resin to dry out. Finally, the product was collected and titrated with HCl (0.1 N) using phenolphthalein as an indicator to determine the hydroxide concentration of the solution.

**General procedure for the synthesis of zeolites in fluoride medium using different OSDAs.** In a polypropylene beaker, an aqueous solution of the corresponding ch-OSDAx with a known concentration was mixed with tetraethylorthosilicate (TEOS) and germanium oxide ($GeO_2$), for Ge-containing samples, and aluminum triisopropoxide ($Al[iPrO]_3$), in the case of the Al-containing samples, following the specified molar ratios in Table 1. The mixture was maintained under mechanical stirring for 12 h. After that, aqueous HF solution was added, and the amount of water was adjusted by evaporation or water addition until the mixture reached the following final composition of $x$ $SiO_2$: $y$ $GeO_2$: $z$ $Al_2O_3$: 0.25 ch-OSDA(OH)$_2$: 0.5 HF: 5 $H_2O$.

The synthesis gel was transferred to Teflon containers, which were then placed in stainless steel autoclaves. The autoclave was maintained at temperatures of 150–175 °C under constant agitation for 3-20 days. The resulting material is recovered by filtration, washed with distilled water, and dried at 100 °C for 6 h, obtaining the zeolite as a powder.

**Synthesis of enantiomorphically pure STW zeolite (S-STW) in the form of germanosilicate in fluoride medium using OSDA3.** In a polypropylene beaker, an aqueous solution of ch-OSDA3(OH)$_2$ (0.11 M, 0.455 mL, 5 mmol), tetraethylorthosilicate (TEOS) (2.77 g, 13.33 mmol) and $GeO_2$ (0.70 g, 6.66 mmol) were mixed under mechanical stirring for 12 h. After evaporation of the ethanol formed, aqueous HF solution (50 wt.%, 400 mg, 10 mmol) was added, and the amount of water was adjusted by evaporation or water addition until the mixture reached the following final composition expressed in molar ratios of 0.66 $SiO_2$: 0.33 $GeO_2$: 0.25 ch-OSDA3(OH)$_2$: 0.5 HF: 5 $H_2O$.

The synthesis gel was transferred to 15 mL Teflon containers, which were then placed in stainless steel autoclaves. The autoclave was kept at 175 °C under constant tumbling for 7 days. The resulting material is recovered by filtration, washed with distilled water (2 L × autoclave), and dried at 100 °C for 6 h, obtaining in STW zeolite as a light brown solid (1.15 g, 66%).

Al-containing samples were synthesized by adding the appropriate amount of aluminum triisopropoxide to the synthesis gel to reach the target Al contents.

**Synthesis of racemic STW zeolite (rac-STW) in the form of germanosilicate in fluoride medium using 2-ethyl-1,3,4-trimethylimidazolium (ETMI).** In a polypropylene beaker, an aqueous solution of ETMI(OH) (0.43 M, 20 mmol), tetraethylorthosilicate (TEOS) (5.54 g, 26.67 mmol) and $GeO_2$ (1.40 g, 13.32 mmol) were mixed under mechanical stirring for 12 h. After evaporation of the ethanol formed, aqueous HF solution (50 wt.%, 800 mg, 20 mmol) was added, and the amount of water was adjusted by evaporation or water addition until the mixture reached the following final composition expressed in molar ratios of 0.66 $SiO_2$: 0.33 $GeO_2$: 0.5 ETMI(OH): 0.5 HF: 4 $H_2O$.

The synthesis gel was transferred to 15 mL Teflon containers, which were then placed in stainless steel autoclaves. The autoclave was kept at 175 °C under constant tumbling for 5 days. The resulting material is recovered by filtration, washed with distilled water (2 L × autoclave) and dried at 100 °C for 6 h, obtaining the STW zeolite as a white solid (2.3 g, 65 %).

## Physicochemical characterization of the materials
Routine powder X-Ray diffraction (PXRD) patterns were collected using a Cubix PANalytical diffractometer CuKα radiation ($\lambda_1 = 1.5406$ Å) at 45 kV and 40 mA in the $2\theta$ range from 2°–40°. The sample 2-STW as powder was calcined in situ under a dry air stream at 600 °C for 8 h and subsequently cooled down at RT in an Anton-Paar XRK-900 reaction chamber attached to an Empyrean X-Ray diffractometer. The experimental details of single crystal XRD data collection and analysis are provided in Suppl. Table 6. The element analysis (EA) of nitrogen, carbon and hydrogen (N, C, H) contents was carried out in a Fisons EA1108 Elemental Analyzer using sulfanilamide as reference standard. The inorganic compositions of the zeolites were determined by dissolving the sample in acid media. The procedure consists in dissolving 50 mg of zeolite was dissolved in 1 mL of an acid solution of 1:1:3 HF (40%): $HNO_3$ (70%): HCl (33%), overnight. The resulting solution was diluted up to 50 mL using high pure water (MilliQ quality). The Si, Ge and Al contents in the resulting solutions were determined by ICP-OES on a ICP Varian 715-ES instrument.

The fluoride contents of some samples were determined by measuring their $^{19}$F-MAS-NMR spectra of the weighted STW samples. The intensities of the observed resonances were integrated and compared to a fluoride standard (in this case, F-containing zeolite BEC).

Bright and dark field optical microscopy images were obtained in a Leica microscope using polarized light under different orientations.

Field Emission Scanning Electron Microscopy images were obtained in a Zeiss Ultra 55 with an accelerating voltage of 1 kV.

Liquid Nuclear Magnetic Resonance (NMR) spectra were recorded on a Bruker DRX-300 spectrometer, and solid-state NMR spectra were obtained on a Bruker AV-400 spectrometer using magic-angle spinning (MAS) techniques at room temperature.

Micromeritics ASAP 2020 and 2420 volumetric devices were used to measure Ar (87 K) and $N_2$ (77 K) gas adsorption on the calcined samples upon outgassing at 673 K overnight. Apparent BET surface area and micropore volume were calculated from the $N_2$ adsorption isotherm following the recommendations given by IUPAC[40], while the micropore distribution was calculated by applying the Hovarth-Kawazoe model to the high-resolution Ar adsorption isotherm.

## Computational methods
The geometry optimizations have been performed using the GULP code[41,42] employing the Ewald method for summation of the long-range Coulombic interactions, and direct summation of the short-range interactions with a cut-off distance of 12 Å. The BFGS (Broyden−Fletcher−Goldfarb−Shanno) technique[43–46] was used as the cell minimization scheme with a convergence criterion of a gradient norm below 0.001 eV Å$^{-1}$. Full optimizations of all the atoms of the system (zeolite + OSDA) have been performed, and also the unit cell parameters were optimized. For the OSDA, the charge distribution has been obtained by means of a charge equilibration method[47].

The force field for the zeolite, which we usually take from Ghysels et al.[48], and the most recent update in Misturini et al.[49], was not needed in this case since we preferred to fix the zeolite to the atomic coordinates and cell parameters found from the XRD determination of the selected crystal of 2-STW sample (Suppl. Table 2). In this case, the full optimization of the 3 OSDA molecules per STW unit cell (60 $SiO_2$) occurs in the exact microporous space that has been experimentally determined. For the zeolite-template interactions, the force field chosen was that by Kiselev et al.[50], whose parameters were taken from Catlow et al.[51].

The intramolecular OSDA interactions were considered according to the force field by Oie et al.[52], that we have used extensively in our group[53]. This force field meets the conditions of simplicity (small number of simple equations to describe the energetic terms), generality (it can be applied to a large number of organic molecules) and accuracy.

The interaction energy of the system upon the occlusion of OSDA molecules in the zeolite micropores is calculated according to Eq.(1).

$$E_{\text{zeo-OSDA}} \simeq \frac{1}{n}\sum_i^n E^{\text{vdW}}\text{zeo-OSDA}_i \qquad (1)$$

where $n$ is the number of OSDA molecules in the zeolite unit cell, and here for the OSDAs considered, $n = 3$. From the optimized geometry, the van der Waals component of the zeo-OSDA interaction is calculated and given as the unique parameter to assess the suitability of the OSDA in STW.

## Data availability
The data that support the findings of this study are available from the corresponding authors upon request. The X-ray crystallographic data for the structures reported in this article have been deposited at the Cambridge Crystallographic Data Center (CCDC) under deposition numbers CCDC-2278869 to CCDC-2278910. The data can be obtained free of charge from The Cambridge Crystallographic Data Center via www.ccdc.cam.ac.uk/structures. Source data are provided in this paper.

## Code availability
GULP is a program for performing a variety of types of simulation on materials that can be obtained from Curtin University upon request (https://gulp.curtin.edu.au).

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

## Acknowledgements
We thank SGAI-CSIC and ASIC-UPV for the use of computational facilities. RIAIDT-USC analytical facilities for the SCXRD analyses, and the Electron Microscopy Service of UPV are also acknowledged. Financial support by the Spanish Ministry of Science and Innovation (CEX2021–001230-S and PID2022-136934OB-100 grants funded by MCIN/AEI/10.13039/ 501100011033 funded by "ERDF A way of making Europe" and TED2021–130191B-C41 grant funded by the European Union NextGenerationEU/PRTR) are gratefully acknowledged. The authors also thank the financial support of the Generalitat Valenciana (Prometeo 2021/077). A.S. thanks for the grant BES-2016–078684. This study forms part of the Advanced Materials program and was supported by MCIN with partial funding from European Union Next Generation EU (PRTR-C17. I1) and by Generalitat Valenciana (MFA/2022/012 and MFA/2022/047).

## Author contributions
A.S., F.R and S.V. conceived the project. F.R and S.V. supervised the project. A.S. performed the organic and inorganic synthesis work, as well as some of the physicochemical characterization. A.L.L.-S. carried out the single-crystal X-ray diffraction data collection and performed the corresponding analyses. J.L.J. performed Rietveld refinements of powder X-ray diffraction data. G.S. performed the computational calculations. All the authors discussed the results participated in the writing of the original draft and revised the final manuscript.

## Competing interests
The authors declare no competing interests.
