## [Peer Review File · Nature Communications]

Sugar-based synthesis of an enantiomorphically pure zeoliteREVIEWER COMMENTS

Reviewer #1 (Remarks to the Author):

The work described in this manuscript is very good. The topic of chiral molecular sieves is appropriate for Nature Communications, and the results of sufficient significance to merit publication. I have a few comment to hopefully assist in constructing the final version of the manuscript.

1. There is really no way to prove that a powder is enantiomerically pure. The authors have done a good job towards showing that their samples have very high ee's. Analysis of 30 crystals is an excellent attempt. If Nature Communications wants to be absolutely correct on this, I would suggest that "enantiomerically pure" be put in quotes and then somewhere in the manuscript the authors discuss that by analyses of 30 crystals the ee must be in the high 90's. Therefore, they are calling this high ee "enantiomerically pure."

2. The samples are prepared at low Si/Ge. This leads to low stability of the material. Note that the calcination was performed in dry air. Somewhere in the presentation, this limitation should be mentioned. It does not appear that this new organic structure directing agent provides for materials that can be synthesized at compositions significantly difference than previously published.

3. How were the compositions measured? There needs to be at least one sample that has chemical composition measured independently from the structural analyses. What is F content and what is the ion balance from a bulk elemental analysis?

4. There is no such thing as a BET surface area for a material like this. There must be multi-layered adsorption to use the BET correctly. A lot of people in this field misuse the BET equation. I suggest they remove BET surface area numbers as they are meaningless. It is appropriate to use pore volumes and they do have those listed already.

5. It would have been nice to see some type of functional expression of the chirality. For example adsorption and/or catalysis. However, I believe that the structural work that has been accomplished is sufficient for publication in Nature Communications.

Overall, this is a nice piece of work.

Reviewer #2 (Remarks to the Author):

The manuscript describes a very interesting chiral OSDA for the preparation of enantiomerically-pure STW chiral zeolite, using a sugar-based cation derived from isomannide. The work is of great interest since it is the first clear demonstration of the crystallization of an enantiomerically-pure zeolite by using this particular chiral OSDAs. Although previous reports have shown at least enantio-enriched chiral zeolites with STW, CZP or ITV frameworks, this is the first clear proof of the enantiopurity of a chiral zeolite. A limitation, however, both in terms of characterization and application, is given by the fact that only one enantiomer of the zeolite, that imposed by the absolute configuration of the chiral sugar used as precursor, can be obtained (S-STW). Although the work is of high interest and correctly performed, with conclusions clearly supported by the data, however a number of issues need to be addressed:

- The main question relates to the use of these new zeolites for asymmetric catalysis, as was done by Brand and coworkers in their original work about STW (ref. 20), where it was reported the asymmetric catalytic activity of STW chiral zeolites for the ring-opening of chiral linear epoxides with methanol or even larger alcohols (ref. 20 and ref. 2), as well as for the enantioselective adsorption of 2-butanol. Opposite chiral behaviors observed for both zeolite enantiomers clearly evidenced the asymmetric activity of the chiral zeolites. In this regard, have the authors tried their enantiopure chiral zeolite for asymmetric catalysis and adsorption of chiral substrates processes? The authors should comment on this on the manuscript, since this is a crucial issue.

- The authors analyze by computational methods the fit of the different cations (OSDA1 to 4) in the STW zeolite (Table S1 and Figure S3): with which STW polymorph (P6122 or P6322) were the calculations with the cations performed? In this regard, it would be highly interesting to study the interaction energies, in particular of OSDA3, in both STW crystalline polymorphs (P6122 and P6322) in order to see the distinct fit of each enantiomer of the organic cation in both crystalline zeolite polymorphs (similarly as was done by Brand et al. in ref. 20). Analysis of the different chiral host-guest match of both diastereomeric pairs would be very interesting to understand the transfer of the chirality of the ch-OSDA into the chiral zeolite growing; this information would add great value to the manuscript.
- The absolute configuration of the crystals of the STW zeolite are clearly determined by single-crystal X-Ray Diffraction. Apart from the use of optical microscopy (Figure S9), have the authors considered the use of circular dichroism to characterize their samples? The material would not be active in the usual Electronic Circular Dichroism, but should be active in Vibrational Circular Dichroism, and if so, this could provide a fingerprint to characterize the handedness of STW chiral zeolites when single-crystal studies are not possible.
- The authors report that: "An important feature of isomannide is the absence of a specular analogue, rendering as a highly appropriate starting molecule for the synthesis of new enantiopure ch-OSDAs". However, as previously mentioned, this could represent a disadvantage since only one handedness of the zeolite, that imposed by nature, would be available for potential applications.
- In the ¹³C NMR spectra (Figure S5) of the zeolite, bands corresponding to C4 and C5 split. What is the reason for such splitting? Is it because of a different environment of the corresponding C atoms (two Cs for each)? This might be related to the asymmetric position of the ch-OSDA within the zeolite, as determined from Rietveld, and this information would be interesting to understand the transfer of chirality into the zeolite.
- Si-STW has not been obtained with this OSDA, according to the authors. Have they tried to use seeds of silicogermanate-STW in order to favor the crystallization of Si-STW, or even to achieve smaller crystals more appropriate for applications?
- Given the very limited number of references related to enantio-enriched chiral zeolites, some citations are missed: for instance, the first report of an enantiomerically-enriched zeolite by using nucleotides (derived from chiral sugars) (Zhang et al., Nucleotide-catalyzed conversion of racemic zeolite-type zincophosphate into enantioenriched crystals, *Angew. Chem. Int. Ed.* 2009, 48, 6049–6051). On the other hand, more recent publications by de la Serna et al. have reached higher ee's than those mentioned in the current manuscript, of up to 55 % (de la Serna et al., Inversion of chirality in GTM-4 enantio-enriched zeolite driven by a minor change of the structure-directing agent, *Chem. Commun.* 2022, 58, 13083; de la Serna et al., Asymmetric catalysis within chiral zeolitic nanopores: Chiral host-guest match in GTM-3 zeolite, *Catal. Today* 2024, 426, 114389).

Reviewer #3 (Remarks to the Author):

Sala et al report the synthesis of an enantiomorphically pure germanosilicate STW zeolite by using a sugar-derived chiral-organic structure-directing agent. The materials were characterized by powder and single-crystal x-ray diffraction, optical microscopy, scanning electron microscopy, NMR, and gas adsorption measurements. The evidence of the synthesis of an enantiomorphically pure STW zeolite was presented. The enantiomerically enriched STW germanosilicate zeolite has been reported previously. The novelty of this manuscript is to provide a new chiral-organic structure-directing agent for synthesizing enantiomorphically pure STW zeolite, which merits its publication in *Nat. Commun.* However, the following comments need to be addressed before its acceptance for publication.

1. In the Abstract and Conclusions, the authors stated that they used a "novel" or "innovative" zeolite synthesis approach to synthesize enantiomerically pure S-STW zeolite. The synthesis of chiral zeolite with a chiral OSDA is a known method. The statement is overstated.
2. The application of the enantiomorphically pure STW zeolite in chiral separation or catalysis was not provided. This makes the work incomplete.
3. CD spectra should be provided as additional evidence for enantiomorphically pure zeolite.
4. The graphs in Figs 2 and 3 should be labeled and annotated.
5. Fig. 2 and Fig. S9 clearly show that the STW samples are not pure, contaminated with amorphous materials. There are also small rod-like crystals in Fig S9 which are different with the STW crystals. What are they? These make the claim "The purity of the S-STW sample was

confirmed by carrying out the Rietveld refinement" suspicious. In Fig. 4 there are also some peaks not due to the STW zeolite. Please explain.

RESPONSE TO REVIEWERS' COMMENTS

Reviewer #1 (Remarks to the Author):

The work described in this manuscript is very good. The topic of chiral molecular sieves is appropriate for Nature Communications, and the results of sufficient significance to merit publication. I have a few comment to hopefully assist in constructing the final version of the manuscript.

1. There is really no way to prove that a powder is enantiomerically pure. The authors have done a good job towards showing that their samples have very high ee's. Analysis of 30 crystals is an excellent attempt. If Nature Communications wants to be absolutely correct on this, I would suggest that "enantiomerically pure" be put in quotes and then somewhere in the manuscript the authors discuss that by analyses of 30 crystals the ee must be in the high 90's. Therefore, they are calling this high ee "enantiomerically pure."

The referee is correct in noting that it's challenging to determine absolute enantiomeric purity. However, the use of standards labelled as 'enantiomerically pure' as a reference point for 100% purity is widely spread in determining chirality's. We thank to the referee for appreciating our effort in analysing 30 crystals, which is a good start. This statement has been incorporated into the main manuscript in page 6, lines 15 to 18.

However, to be 95% confident that our sample is over 95% enantiomerically pure, we would need to analyse at least 60 crystals. And if we want even higher purity or confidence levels, we would need to analyse hundreds of crystals, which is simply not feasible for most of the laboratories, including ours.

2. The samples are prepared at low Si/Ge. This leads to low stability of the material. Note that the calcination was performed in dry air. Somewhere in the presentation, this limitation should be mentioned. It does not appear that this new organic structure directing agent provides for materials that can be synthesized at compositions significantly difference than previously published.

The purpose of employing the sugar-based organic structure-directing agent (OSDA) for the synthesis of Ge-STW is not primarily enhancing the stability of the calcined zeolite. Rather, it serves as an effective method of transmitting its chirality to the resulting solid material. This novel OSDA does not alter the chemical composition range of the inorganic framework of STW zeolites. Consequently, the referee's observation regarding the instability of calcined Ge-STW upon exposure to atmospheric moisture is valid as it was stated in the previous manuscript (in the new version of the manuscript in page 4, lines 36 to 38. Nevertheless, we have also included this clarification in the updated manuscript in the figure caption of Figure 4 of the main text.

3. How were the compositions measured? There needs to be at least one sample that has chemical composition measured independently from the structural analyses. What is F content and what is the ion balance from a bulk elemental analysis?

The samples Ge-STW described in this work were analysed by ICP (see table S2 that report the inorganic compositions). The C, H, N and F contents were determined in some samples. The analytical procedures were as follows:

50 mg of zeolite were dissolved in 1 mL of an acid solution of 1:1:3 HF (40%): HNO₃ (70%): HCl (33%), overnight. The resulting solution was diluted up to 50 mL using high pure water (MilliQ quality). The Si and Ge contents in the resulting solution were determined by ICP-OES on a ICP Varian 715-ES instrument.

C, H and N were determined on a EuroEA300 instrument from Eurovector.

The Fluoride contents of some samples were determined by measuring their ¹⁹F-MAS-NMR spectra of the weighted STW samples. The intensities of the observed resonances were integrated and compared to a Fluoride standard (in this case, F-containing zeolite BEC). The Fluoride contents were in all cases between 1.8 to 1.9 wt%, in good agreement to that obtained from single-crystal diffraction data (i.e. 6F/U.C).

These paragraphs have been included in the experimental section in page 15, lines 7 to 14 of the main manuscript.

4. There is no such thing as a BET surface area for a material like this. There must be multi-layered adsorption to use the BET correctly. A lot of people in this field misuse the BET equation. I suggest they remove BET surface area numbers as they are meaningless. It is appropriate to use pore volumes and they do have those listed already.

The referee's concern regarding the BET-area values is acknowledged. However, it's important to note that these values are widely utilized in scientific literature. Additionally, the International Union of Pure and Applied Chemistry (IUPAC) recognizes their use as an apparent surface area, considering them as a valuable adsorbent "fingerprint" (new reference 40 in the manuscript). To address this, we have amended the terminology from 'BET area' or 'BET surface area' to 'apparent BET surface area', with the relevant reference being duly incorporated in page 15, line 24 of the main manuscript and in the table S3 in Supplementary materials.

5. It would have been nice to see some type of functional expression of the chirality. For example adsorption and/or catalysis. However, I believe that the structural work that has been accomplished is sufficient for publication in Nature Communications. Overall, this is a nice piece of work.

Thank you very much for your comment. We attempted adsorption experiments with S-2-butanol and R-2-butanol on the 2-STW sample, following the findings from Davis et al., who reported preferential adsorption of S-2-butanol on enriched S-STW zeolite (and vice versa for R-2-butanol). However, we did not observe any vapor uptake on our sample. To verify the retention of microporosity, we measured the CO₂ adsorption isotherm before and after the alcohol adsorption experiments, being both identical and therefore, confirming that there is no loss of microporosity during the adsorption experiments. The lack of alcohol adsorption may be attributed to the large crystal size of our zeolite, necessary for single crystal structural elucidation, but hindering diffusion through the helicoidal channel of the STW sample.

Additionally, catalytic experiments were performed to investigate epoxide ring-opening reactions utilizing aluminum-containing STW samples (named as Al-S-STW) as catalysts. Unfortunately, the results of these experiments demonstrated notably low epoxide conversion

rates across various chain lengths, as outlined in the table below (provided solely for referee evaluation purposes).

Material	epoxide	Conversion (%)	Selectivity to A (%)	Selectivity to B (%)
Al-S-STW	1,2-epoxybutane	6	67 (e.e.: 0)	33 (e.e.: 2.5)
Al-S-STW	1,2-epoxyhexane	2	49 (e.e.: 4)	51 (e.e.: 3.6)
FAU-15	1,2-epoxybutane	94	46 (e.e.: 0)	54 (e.e.: 0)

Reaction temperature: 25°C, 7 mmol epoxide, 14 mmol i-propanol, 30 mg A-S-STW, reaction time: 1h

This table shows that the achieved conversion for 1,2-epoxyhexane ring-opening was 2%, with an approximate 1:2 ratio of product A to product B., with an enantiomeric excess ranging between 4 to 3.5% in the resultant products. When evaluating shorter alkyl chain epoxides, such as 1,2-epoxybutane, a higher conversion rate of 6% was achieved, albeit with enantiomeric excesses in the products ranging from 1.5 to 2.5%.

When compared with the results obtained using another zeolite with a larger pore size, we observed a significant increase in conversion, reaching nearly quantitative values. This supports our hypothesis that poor diffusion of reactants through the smaller pores of the material limits reaction efficiency.

This limitation in activity and selectivity can be attributed to the very large crystal size of the catalysts, hampering the diffusion of reactants and/or products through the helicoidal pores during the reaction, and therefore the limited activity is mostly occurring at the external surface of the crystals of zeolite STW.

Despite efforts to reduce the crystal size of the S-STW materials to avoid these limitations, we did not observe a significant decrease in crystal size for well-crystallized solids. **These limitations have been acknowledged in page 8, lines 19 to 30 of the revised version of the manuscript.**

Reviewer #2 (Remarks to the Author):

The manuscript describes a very interesting chiral OSDA for the preparation of enantiomerically-pure STW chiral zeolite, using a sugar-based cation derived from isomannide. The work is of great interest since it is the first clear demonstration of the crystallization of an enantiomerically-pure zeolite by using this particular chiral OSDAs. Although previous reports have shown at least enantio-enriched chiral zeolites with STW, CZP or ITV frameworks, this is the first clear proof of the enantiopurity of a chiral zeolite.

A limitation, however, both in terms of characterization and application, is given by the fact that only one enantiomer of the zeolite, that imposed by the absolute configuration of the chiral sugar used as precursor, can be obtained (S-STW). Although the work is of high interest and correctly performed, with conclusions clearly supported by the data, however a number of issues need to be addressed:

- The main question relates to the use of these new zeolites for asymmetric catalysis, as was done by Brand and coworkers in their original work about STW (ref. 20), where it was reported the asymmetric catalytic activity of STW chiral zeolites for the ring-opening of chiral linear epoxides with methanol or even larger alcohols (ref. 20 and ref. 2), as well as for the enantioselective adsorption of 2-butanol. Opposite chiral behaviors observed for both zeolite enantiomers clearly evidenced the asymmetric activity of the chiral zeolites. In this regard, have the authors tried their enantiopure chiral zeolite for asymmetric catalysis and adsorption of chiral substrates processes? The authors should comment on this on the manuscript, since this is a crucial issue.

Thank you very much for your comment. We attempted adsorption experiments with S-2-butanol and R-2-butanol on the 2-STW sample, following the findings from Davis et al., who reported preferential adsorption of S-2-butanol on enriched S-STW zeolite (and vice versa for R-2-butanol). However, we did not observe any vapor uptake on our sample. To verify the retention of microporosity, we measured the CO₂ adsorption isotherm before and after the alcohol adsorption experiments, being both identical and therefore, confirming that there is no loss of microporosity during the adsorption experiments. The lack of alcohol adsorption may be attributed to the large crystal size of our zeolite, necessary for single crystal structural elucidation, but hindering diffusion through the helicoidal channel of the STW sample.

Additionally, catalytic experiments were performed to investigate epoxide ring-opening reactions utilizing aluminum-containing STW samples (named as Al-S-STW) as catalysts. Unfortunately, the results of these experiments demonstrated notably low epoxide conversion rates across various chain lengths, as outlined in the table below (provided solely for referee evaluation purposes).

Material	epoxide	Conversion (%)	Selectivity to A (%)	Selectivity to B (%)
Al-S-STW	1,2,-epoxybutane	6	67 (e.e.: 0)	33 (e.e.: 2.5)
Al-S-STW	1,2-epoxyhexane	2	49 (e.e.: 4)	51 (e.e.: 3.6)
FAU-15	1,2-epoxybutane	94	46 (e.e.: 0)	54 (e.e.: 0)

Reaction temperature: 25°C, 7 mmol epoxide, 14 mmol i-propanol, 30 mg A-S-STW, reaction time: 1h

This table shows that the achieved conversion for 1,2-epoxyhexane ring-opening was 2%, with an approximate 1:2 ratio of product A to product B., with an enantiomeric excess ranging between 4 to 3.5% in the resultant products. When evaluating shorter alkyl chain epoxides, such as 1,2-epoxybutane, a higher conversion rate of 6% was achieved, albeit with enantiomeric excesses in the products ranging from 1.5 to 2.5%.

When compared with the results obtained using another zeolite with a larger pore size, we observed a significant increase in conversion, reaching nearly quantitative values. This supports our hypothesis that poor diffusion of reactants through the smaller pores of the material limits reaction efficiency.

This limitation in activity and selectivity can be attributed to the very large crystal size of the catalysts, hampering the diffusion of reactants and/or products through the helicoidal pores during the reaction, and therefore the limited activity is mostly occurring at the external surface of the crystals of zeolite STW.

Despite efforts to reduce the crystal size of the S-STW materials to avoid these limitations, we did not observe a significant decrease in crystal size for well-crystallized solids. **These limitations have been acknowledged in page 8, lines 19 to 30 of the revised version of the manuscript.**

- The authors analyze by computational methods the fit of the different cations (OSDA1 to 4) in the STW zeolite (Table S1 and Figure S3): with which STW polymorph (P6122 or P6322) were the calculations with the cations performed? In this regard, it would be highly interesting to study the interaction energies, in particular of OSDA3, in both STW crystalline polymorphs (P6122 and P6322) in order to see the distinct fit of each enantiomer of the organic cation in both crystalline zeolite polymorphs (similarly as was done by Brand et al. in ref. 20). Analysis of the different chiral host-guest match of both diastereomeric pairs would be very interesting to understand the transfer of the chirality of the ch-OSDA into the chiral zeolite growing; this information would add great value to the manuscript.

We appreciate the reviewer's insightful comment, and indeed, we didn't specify in the manuscript that all our calculations were conducted in the S enantiomorph of STW, corresponding to the pure enantiomorph we synthesized. Consequently, this aspect has been duly incorporated into the updated version as follows:

Main manuscript:

Page 4, line 6 and 7: "only OSDA3 is stabilized in the micropore of this zeolite" has been changed by "only OSDA3 is stabilized in the micropore of S-STW (P6₁22 space group)".

Caption of Figure S3:

"in STW zeolite" has been changed by "in S-STW zeolite (P6₁22 space group)".

Caption of Table S1:

“Calculated STW-OSDA” has been changed by “Calculated S-STW-OSDA”

We would like to stress here that the two enantiomorphs do not belong to the space groups indicated above by the reviewer but rather $P6_122$ and $P6_522$, as indicated in the main manuscript: “In this case, six crystals were indexed in the space group $P6_522$ (enantiomorph R) and five in the space group $P6_122$ (enantiomorph S) indicating that a racemic mixture of both enantiomorphs was formed.”

More importantly, in response to the reviewer's suggestion, we have optimized the location of OSDA3 in R-STW. The results are shown in the new Figure S4 along with corresponding explanatory text in page 4, lines 11 to 13, which evidence that the curvature of OSDA3 does not align well with the helicoidal channel of R-STW. Specifically, our Monte Carlo algorithm required 10 times more cycles (120000) to determine this geometry and could only accommodate one OSDA3 molecule, whereas in the case of S-STW, three OSDA3 molecules could be readily allocated within its micropore system using the same Monte Carlo algorithm.

- The absolute configuration of the crystals of the STW zeolite are clearly determined by single-crystal X-Ray Diffraction. Apart from the use of optical microscopy (Figure S9) have the authors considered the use of circular dichroism to characterize their samples? The material would not be active in the usual Electronic Circular Dichroism, but should be active in Vibrational Circular Dichroism, and if so, this could provide a fingerprint to characterize the handedness of STW chiral zeolites when single-crystal studies are not possible.

We conducted circular dichroism measurements in the UV-visible region. However, the results obtained were inconclusive as dichroic signals were observed at certain angles of incidence, while they were absent at others. For the purpose of review, we have included examples of spectra collected from one of our samples at different angles. In the figure below, a clear dichroic signal at 200 nm is evident at 0 and 180°, while no signal is observed at 90°. This observation may suggest the presence of chirality in the studied solids, although conclusive interpretation needs further investigation, which falls outside our current expertise.

Additionally, we attempted to follow the referee's recommendation of measuring the vibrational circular dichroism of the reported STW materials. This spectroscopic technique is unconventional, and unfortunately, we were unable to locate any physical-chemical service at Spanish universities or Public Research Institutes offering this capability.

Instead, we have relied on single-crystal X-ray diffraction (SCXRD) and the Flack parameter as our primary methodologies. These are widely accepted in the scientific community for determining

the chirality of crystal structures. These methods provide a more direct and unambiguous way to assess the enantiomeric purity of the zeolites we have synthesized, ensuring that our conclusions are robust.

- The authors report that: "An important feature of isomannide is the absence of a specular analogue, rendering as a highly appropriate starting molecule for the synthesis of new enantiopure ch-OSDAs". However, as previously mentioned, this could represent a disadvantage since only one handedness of the zeolite, that imposed by nature, would be available for potential applications.

The referee's comment is correct, as our synthesis route indeed only yields one enantiomorph of the STW zeolite. However, in nature, chiral compounds typically appear in either the R or S configuration, maintaining a high level of purity. Achieving such enantiomeric excess through synthetic routes is challenging, often resulting in impurities of the opposite enantiomer during organic synthesis. Therefore, our approach in this study was to utilize natural chiral products that lack a counterpart chiral enantiomer. This enables the synthesis of pure chiral organic structure-directing agents (OSDAs), as racemization is not possible. Furthermore, the syntheses described in this work are very simple, facilitating the preparation of OSDAs at a multigram scale necessary for studies on new zeolite synthesis.

- In the ^{13}C NMR spectra (Figure S5) of the zeolite, bands corresponding to C4 and C5 split. What is the reason for such splitting? Is it because of a different environment of the corresponding C atoms (two Cs for each)? This might be related to the asymmetric position of the ch-OSDA within the zeolite, as determined from Rietveld, and this information would be interesting to understand the transfer of chirality into the zeolite.

The presence of four distinct configurations of the encapsulated OSDA within pore system of the S-STW zeolite could account for the splitting observed in the ^{13}C -CP-SS-NMR spectrum of the as-synthesized material. The different docking of the organic molecules within the helicoidal channel induces slight modifications in the surrounding environment of the carbon nuclei of the OSDA, leading to the observed signal splitting in the ^{13}C -NMR spectrum. The explanatory text has been included in the figure caption of Figure S5 of the new manuscript.

- Si-STW has not been obtained with this OSDA, according to the authors. Have they tried to use seeds of silicogermanate-STW in order to favor the crystallization of Si-STW, or even to achieve smaller crystals more appropriate for applications?

The use of STW seeds to increase the Si/Ge ratio and/or reduce crystal size has been investigated, albeit with limited success. Seeding with a high concentration of Ge-containing STW zeolite (up to 30% seeding) has led to the formation of amorphous solids when attempting to increase the Si/Ge ratio. In addition, attempts to diminish crystal size by introducing seeds into the synthesis gel were unsuccessful, resulting in the formation of Ge-containing STW zeolites with crystal sizes similar to those of unseeded samples or in the formation of impurities in the final solid. A statement addressing these outcomes has been incorporated accordingly in page 8, lines 19 to 30.

- Given the very limited number of references related to enantio-enriched chiral zeolites, some citations are missed: for instance, the first report of an enantiomerically-enriched zeolite by using nucleotides (derived from chiral sugars) (Zhang et al., Nucleotide-catalyzed conversion of racemic zeolite-type zincophosphate into enantioenriched crystals, *Angew. Chem. Int. Ed.* 2009, 48, 6049–6051). On the other hand, more recent publications by de la Serna et al. have reached higher ee's than those mentioned in the current manuscript, of up to 55 % (de la Serna et al., Inversion of chirality in GTM-4 enantio-enriched zeolite driven by a minor change of the structure-directing agent, *Chem. Commun.* 2022, 58, 13083; de la Serna et al., Asymmetric catalysis within chiral zeolitic nanopores: Chiral host-guest match in GTM-3 zeolite, *Catal. Today* 2024, 426, 114389).

We are grateful to the reviewer for this comment. The missing references have been added to the updated version of the manuscript as new references 11, 21, 25 and 26. All the references have been renumbered.

Reviewer #3 (Remarks to the Author):

Sala et al report the synthesis of an enantiomorphically pure germanosilicate STW zeolite by using a sugar-derived chiral-organic structure-directing agent. The materials were characterized by powder and single-crystal x-ray diffraction, optical microscopy, scanning electron microscopy, NMR, and gas adsorption measurements. The evidence of the synthesis of an enantiomorphically pure STW zeolite was presented. The enantiomerically enriched STW germanosilicate zeolite has been reported previously. The novelty of this manuscript is to provide a new chiral-organic structure-directing agent for synthesizing enantiomorphically pure STW zeolite, which merits its publication in Nat. Commun. However, the following comments need to be addressed before its acceptance for publication.

1. In the Abstract and Conclusions, the authors stated that they used a "novel" or "innovative" zeolite synthesis approach to synthesize enantiomerically pure S-STW zeolite. The synthesis of chiral zeolite with a chiral OSDA is a known method. The statement is overstated.

The reviewer's comment is right, as chiral organic structure-directing agents (OSDAs) have been utilized previously attempting to produce chiral zeolites, albeit with varying degrees of success. Many previous efforts have resulted in non-chiral zeolites, as we also report in this study, while a few have achieved significant enrichments in R and S enantiomorphous zeolites. However, none have been able to claim the synthesis of a 'pure enantiomorph' as we demonstrate in this work.

The terms 'novel' and 'innovative' in our study specifically refer to the utilization of sugars as chiral building blocks, as emphasized in the title, to synthesize unique chiral organic structure-directing agents (OSDAs). These newly developed chiral compounds exhibited remarkable selectivity in imparting their chirality to the zeolite. To our knowledge, the approach of employing sugar-derived OSDAs has not been previously explored in the open literature or patented methods. Furthermore, this innovative strategy has led to the synthesis of enantiomorphically pure zeolite, thereby enhancing its novelty. We firmly believe that these aspects justify characterizing our work as 'novel' and/or 'innovative'. This discovery may pave the way for a new research direction in the synthesis of chiral porous solids.

2. The application of the enantiomorphically pure STW zeolite in chiral separation or catalysis was not provided. This makes the work incomplete.

Thank you very much for your comment. We attempted adsorption experiments with S-2-butanol and R-2-butanol on the 2-STW sample, following the findings from Davis et al., who reported preferential adsorption of S-2-butanol on enriched S-STW zeolite (and vice versa for R-2-butanol). However, we did not observe any vapor uptake on our sample. To verify the retention of microporosity, we measured the CO₂ adsorption isotherm before and after the alcohol adsorption experiments, being both identical and therefore, confirming that there is no loss of microporosity during the adsorption experiments. The lack of alcohol adsorption may be attributed to the large crystal size of our zeolite, necessary for single crystal structural elucidation, but hindering diffusion through the helicoidal channel of the STW sample.

Additionally, catalytic experiments were performed to investigate epoxide ring-opening reactions utilizing aluminum-containing STW samples (named as Al-S-STW) as catalysts. Unfortunately, the results of these experiments demonstrated notably low epoxide conversion

rates across various chain lengths, as outlined in the table below (provided solely for referee evaluation purposes).

Material	epoxide	Conversion (%)	Selectivity to A (%)	Selectivity to B (%)
Al-S-STW	1,2-epoxybutane	6	67 (e.e.: 0)	33 (e.e.: 2.5)
Al-S-STW	1,2-epoxyhexane	2	49 (e.e.: 4)	51 (e.e.: 3.6)
FAU-15	1,2-epoxybutane	94	46 (e.e.: 0)	54 (e.e.: 0)

Reaction temperature: 25°C, 7 mmol epoxide, 14 mmol i-propanol, 30 mg A-S-STW, reaction time: 1h

This table shows that the achieved conversion for 1,2-epoxyhexane ring-opening was 2%, with an approximate 1:2 ratio of product A to product B., with an enantiomeric excess ranging between 4 to 3.5% in the resultant products. When evaluating shorter alkyl chain epoxides, such as 1,2-epoxybutane, a higher conversion rate of 6% was achieved, albeit with enantiomeric excesses in the products ranging from 1.5 to 2.5%.

When compared with the results obtained using another zeolite with a larger pore size, we observed a significant increase in conversion, reaching nearly quantitative values. This supports our hypothesis that poor diffusion of reactants through the smaller pores of the material limits reaction efficiency.

This limitation in activity and selectivity can be attributed to the very large crystal size of the catalysts, hampering the diffusion of reactants and/or products through the helicoidal pores during the reaction, and therefore the limited activity is mostly occurring at the external surface of the crystals of zeolite STW.

Despite efforts to reduce the crystal size of the S-STW materials to avoid these limitations, we did not observe a significant decrease in crystal size for well-crystallized solids. **These limitations have been acknowledged in page 8, lines 19 to 30 of the revised version of the manuscript.**

3. CD spectra should be provided as additional evidence for enantiomorphically pure zeolite.

We conducted circular dichroism measurements in the UV-visible region. However, the results obtained were inconclusive as dichroic signals were observed at certain angles of incidence, while they were absent at others. For the purpose of review, we have included examples of spectra collected from one of our samples at different angles. In the figure below, a clear dichroic signal at 200 nm is evident at 0 and 180°, while no signal is observed at 90°. This observation may suggest the presence of chirality in the studied solids, although conclusive interpretation needs further investigation, which falls outside our current expertise.

Additionally, we attempted to measure the vibrational circular dichroism of the reported STW materials. This spectroscopic technique is unconventional, and unfortunately, we were unable to locate any physical-chemical service at Spanish universities or Public Research Institutes offering this capability.

Instead, we have relied on single-crystal X-ray diffraction (SCXRD) and the Flack parameter as our primary methodologies. These are widely accepted in the scientific community for determining the chirality of crystal structures. These methods provide a more direct and unambiguous way to assess the enantiomeric purity of the zeolites we have synthesized, ensuring that our conclusions are robust and scientifically sound.

4. The graphs in Figs 2 and 3 should be labeled and annotated.

Thanks to the reviewer for this comment. The figures 2 and 3 have been modified according to the referee's indications.

5. Fig. 2 and Fig. S9 clearly show that the STW samples are not pure, contaminated with amorphous materials. There are also small rod-like crystals in Fig S9 which are different with the STW crystals. What are they? These make the claim "The purity of the S-STW sample was confirmed by carrying out the Rietveld refinement" suspicious. In Fig. 4 there are also some peaks not due to the STW zeolite. Please explain.

SEM images in Figure 2 show several preparations of S-STW-2. The top-left image demonstrates the homogeneity in crystal size, while the remaining images provide a closer view of the crystals, revealing minor impurities due to sample preparation. The presence of breakage in S-STW crystals, as shown in the bottom right image, may account for other observed 'impurities' in the bottom-left images. Additionally, residues of carbon ribbon are observed in the top-right image, as the samples were mounted on double-sided adhesive carbon tape for SEM measurements.

The referee's observation regarding the rod-like crystals in Figure S10 (in previous manuscript, Fig. S9) is correct. We collected one of these rod-like crystals and obtained its single crystal X-ray diffraction data, revealing a unit cell consistent with quartz-like GeO₂. However, it's important to note that this impurity appeared only in this particular preparation and was not evident in all the other syntheses of S-STW-2 reported in this work. The number of rod-like crystals is significantly lower than those exhibiting the characteristic hexagonal bipyramid shape of STW, and their crystal size is several orders of magnitude smaller. Therefore, the content of this impurity is negligible for most common characterization techniques.

Rietveld refinements further support the very high purity of the materials obtained from gels with a Si/Ge ratio of 2. Figure 4 displays the experimental X-ray pattern and calculated profile,

along with the corresponding difference. The residuals of the fitting are minimal, indicating the absence of any detectable phase other than S-STW.

Regarding the weak and broad diffraction peak observed between 6.5 and 7 degrees in Figure 4, this peak is attributed to the high-temperature chamber XRK900 used for the 'in-situ' calcination of the S-STW-2 material. This 'fake' diffraction is also evident in several of our previous publications where the same attachment was utilized (see for instance Fig S7 and S9 in Chemical Communications 2012, **48**, 215-217 and Fig S12 in Science, 2017, **358**, 1068-1071 among others). Thus, the 'fake' peak can be safely treated as part of the background.

In conclusion, we emphasise that the S-STW-2 samples prepared in this work exhibit very high purity and crystallinity.

REVIEWER COMMENTS

Reviewer #1 (Remarks to the Author):

The authors have done a reasonable job of revising the manuscript. I have a few suggestions for improvement. First, at the end of the new statement that they added on page 6 lines 15-18, I would add something like "and we define here this to represent enantiomorphically pure". Second, the complete composition needs to be provided in the manuscript. The authors now say that they measure F by NMR. Somewhere in the main text, the unit cell composition (Si,Ge,organic,F) must be provided.

It is unfortunate that their attempts at showing function(adsorption or catalysis) were not successful. This would have enhanced the presentation. However, I wonder if some portion of the results they show in the responses to the reviewers should be placed in the SI. Those data are indicative of the samples obtained with this synthetic method.

Reviewer #2 (Remarks to the Author):

The authors have properly addressed all the issues raised by this reviewer. Despite the adsorption/catalysis results were unsuccessful, still the synthesis and characterization work of this new enantiopure zeolite deserves publication. The manuscript is ready for publication in its present form.

Reviewer #3 (Remarks to the Author):

The authors addressed most of my concerns in the revised manuscript. There are two minor issues that should be addressed before its acceptance for publication.

1. The authors should note in Figure S10 that the rod-like crystals are impurities of quartz-like GeO₂.
2. The low catalytic activity for epoxide ring-opening reactions and the adsorption capacity of chiral alcohols on the zeolites S-STW were attributed to the significant diffusional limitations arising from the large crystal size of the synthesized materials. The statement is debatable as other factors such as low crystallinity (due to hydrolysis of the calcined Ge-rich STW), hydrophobicity and low acidity could also have contributions (sometimes may be a more dominant factor). For instance, since the adsorption is usually measured at equilibrium, the diffusion limitation owing to the particle size could be largely overcome. It might not be appropriate to attribute the low adsorption to the large crystal size limitation. However, if the measured zeolite is hydrophobic (i.e. Ge-Si STW), it might not adsorb polar 2-butanol.

RESPONSE TO REVIEWERS' COMMENTS

Reviewer #1 (Remarks to the Author):

The authors have done a reasonable job of revising the manuscript.

Thanks to the reviewer for his/her understanding and helpful comments and suggestions.

I have a few suggestions for improvement. First, at the end of the new statement that they added on page 6 lines 15-18, I would add something like "and we define here this to represent enantiomorphically pure".

The following sentence has been added in the new version of the manuscript: '*This has been taken as the definition of enantiomorphically pure S-STW zeolite in this study.*' (lines 18-19 in page 6 of the new manuscript).

Second, the complete composition needs to be provided in the manuscript. The authors now say that they measure F by NMR. Somewhere in the main text, the unit cell composition (Si,Ge,organic,F) must be provided.

The experimental chemical composition of the unit cell obtained by combining different chemical analyses (ICP, EA and F-NMR data) has been included in line 31 in page 6 of the new version of the manuscript and as footnote of Table S2.

It is unfortunate that their attempts at showing function (adsorption or catalysis) were not successful. This would have enhanced the presentation. However, I wonder if some portion of the results they show in the responses to the reviewers should be placed in the SI. Those data are indicative of the samples obtained with this synthetic method.

The adsorption results of R 2-butanol and S-2-butanol on enantiomorphically pure Ge-STW and the racemic all-silica STW have been included and briefly discussed in the new Figure S12 of supplementary material. The catalytic data on the epoxide ring aperture of 1-epoxibutane and 1-epoxihexane on Al containing-S-STW and a large pore zeolite (FAU-15) have been included as Table S9 in the new version of the supplementary material. The catalytic results are briefly discussed in the footnote of Table S9.

Reviewer #2 (Remarks to the Author):

The authors have properly addressed all the issues raised by this reviewer. Despite the adsorption/catalysis results were unsuccessful, still the synthesis and characterization work of this new enantiopure zeolite deserves publication. The manuscript is ready for publication in its present form.

Thanks to the referee for his/her understanding and the previous helpful comments and suggestions.

Reviewer #3 (Remarks to the Author):

The authors addressed most of my concerns in the revised manuscript.

Thanks to the referee for his/her understanding and helpful comments and suggestions.

There are two minor issues that should be addressed before its acceptance for publication.

1. The authors should note in Figure S10 that the rod-like crystals are impurities of quartz-like GeO₂.

The presence of minor impurities of quartz-like GeO₂ has been mentioned in the figure caption of Figure S10 (see page 11 of supplementary material).

2. The low catalytic activity for epoxide ring-opening reactions and the adsorption capacity of chiral alcohols on the zeolites S-STW were attributed to the significant diffusional limitations arising from the large crystal size of the synthesized materials. The statement is debatable as other factors such as low crystallinity (due to hydrolysis of the calcined Ge-rich STW), hydrophobicity and low acidity could also have contributions (sometimes may be a more dominant factor). For instance, since the adsorption is usually measured at equilibrium, the diffusion limitation owing to the particle size could be largely overcome. It might not be appropriate to attribute the low adsorption to the large crystal size limitation. However, if the measured zeolite is hydrophobic (i.e. Ge-Si STW), it might not adsorb polar 2-butanol.

We have carried out CO₂ isotherm measurements both prior to and after adsorption experiments of R-2-butanol and S-2-butanol to evaluate the accessibility of the micropore volume within the Ge-containing S-STW sample. Our results show that approximately 80% of the CO₂ adsorption capacity remained intact after the series of alcohol adsorption experiments, indicating the preservation of a significant portion of the microporosity within the enantiomorphically pure Ge-STW sample.

Analysis of the kinetic data for each point on the 2-butanol isotherms shows that the uptakes are far from equilibrium, even after a long equilibration period of 2 hours per data point. Extending the equilibration time further would render the measurement of 2-butanol isotherms practically unfeasible within reasonable timeframes. This observation is further supported by the 2-butanol isotherms obtained on pure silica STW with significantly smaller crystal sizes (ranging from 0.5 to 5 microns), which exhibit uptake levels of around 15 wt.% and rapid equilibration, indicative of minimal diffusional limitations. In contrast, the markedly larger crystal size of 70 microns in the S-STW sample clearly introduces diffusional limitations for 2-butanol.

Nevertheless, we can not rule out that other factors may also influence the performance of Ge-containing S-STW material. In our revised manuscript, we have included additional adsorption data (see Figure S12) and catalytic data (see Table S9) to provide further insight into this matter.

REVIEWERS' COMMENTS

Reviewer #1 (Remarks to the Author):

The manuscript is ready for publication.

Reviewer #3 (Remarks to the Author):

In this revised manuscript, the authors have done a nice job of addressing the comments from the previous review. Therefore, I recommend for publication of this work.